# Immunosuppression as a Risk Factor for De Novo Angiotensin II Type Receptor Antibodies Development after Kidney Transplantation

**DOI:** 10.3390/jcm10225390

**Published:** 2021-11-18

**Authors:** Bogdan Marian Sorohan, Ioanel Sinescu, Dorina Tacu, Cristina Bucșa, Corina Țincu, Bogdan Obrișcă, Andreea Berechet, Ileana Constantinescu, Ion Mărunțelu, Gener Ismail, Cătălin Baston

**Affiliations:** 1Department of General Medicine, Carol Davila University of Medicine and Pharmacy, 020021 Bucharest, Romania; umfisinescu@gmail.com (I.S.); obriscabogdan@yahoo.com (B.O.); ileana.constantinescu@imunogenetica.ro (I.C.); gener732000@yahoo.com (G.I.); drcbaston@gmail.com (C.B.); 2Department of Nephrology, Fundeni Clinical Institute, 022328 Bucharest, Romania; andreea.berechet93@yahoo.com; 3Center for Uronephrology and Kidney Transplantation, Fundeni Clinical Institute, 022328 Bucharest, Romania; ticu_dorina@yahoo.com (D.T.); cristinabucsa@yahoo.com (C.B.); tincucorina@gmail.com (C.Ț.); 4Department of Immunogenetics, Fundeni Clinical Institute, 022328 Bucharest, Romania; ionutz.marion@gmail.com

**Keywords:** kidney transplant, angiotensin II type 1 receptor antibody, immunosuppression, induction, maintenance, de novo, risk factor, ATG, tacrolimus

## Abstract

(1) Background: Angiotensin II type I receptor antibodies (AT1R-Ab) represent a topic of interest in kidney transplantation (KT). Data regarding the risk factors associated with de novo AT1R-Ab development are lacking. Our goal was to identify the incidence of de novo AT1R-Ab at 1 year after KT and to evaluate the risk factors associated with their formation. (2) Methods: We conducted a prospective cohort study on 56 adult patients, transplanted between 2018 and 2019. Recipient, donor, transplant, treatment, and complications data were assessed. A threshold of >10 U/mL was used for AT1R-Ab detection. (3) Results: De novo AT1R-Ab were observed in 12 out of 56 KT recipients (21.4%). The median value AT1R-Ab in the study cohort was 8.5 U/mL (inter quartile range: 6.8–10.4) and 15.6 U/mL (10.8–19.8) in the positive group. By multivariate logistic regression analysis, induction immunosuppression with anti-thymocyte globulin (OR = 7.20, 95% CI: 1.30–39.65, *p* = 0.02), maintenance immunosuppression with immediate-release tacrolimus (OR = 6.20, 95% CI: 1.16–41.51, *p* = 0.03), and mean tacrolimus trough level (OR = 2.36, 95% CI: 1.14–4.85, *p* = 0.01) were independent risk factors for de novo AT1R-Ab at 1 year after KT. (4) Conclusions: De novo AT1R-Ab development at 1 year after KT is significantly influenced by the type of induction and maintenance immunosuppression.

## 1. Introduction

The non- human leukocyte antigen (non-HLA) system is an emerging field in kidney transplantation (KT) [1]. Several antigens recognized by non-HLA antibodies have been described in the last 15 years [2,3]. Angiotensin II type I receptor antibodies (AT1R-Ab) have generated the most interest in the KT community amongst the non-HLA antibodies [4]. The frequency of AT1R-Ab in KT ranges from 2.1% to 59%, that of preformed AT1R-Ab being identical to the aforementioned range and that of de novo AT1R-Ab varying between 3.8–51.5% [5]. Study design, immunological risk of the KT recipients, immunosuppression (IS), time of antibody assessment and positivity threshold for detection could be among the factors that influences AT1R-Ab variability [5]. AT1R-Ab are considered autoantibodies and seem to determine graft injury in a complement-independent manner [6,7]. The mechanism of AT1R-Ab formation is complex and involves a chronological sequence of pathophysiological events, beginning with endothelial injury, and continuing with receptor exposure as a neoantigen, stimulation of indirect recognition pathway and stimulation of autoreactive T and B cells for antibody production. This process also includes defining events, such as: stimulation of T helper 17 differentiation, epitope spreading, cross-reactivity, loss of self-tolerance, and an interplay between allo- and autoimmunity [1,2,5,8]. Agents involved in endothelial injury and antibody formation could be different for de novo compared to preformed AT1R-Ab. Even though some studies have shown an association between AT1R-Ab, different rejection phenotypes, negative graft function and survival outcomes, little is known about the risk factors associated with AT1R-Ab development [6,9,10,11,12,13,14,15,16,17]. We sought to determine the incidence of de novo AT1R-Ab at 1 year after KT and the risk factors associated with their formation.

## 2. Materials and Methods

### 2.1. Patients and Study Design

We performed a prospective study on 56 adult patients who underwent a KT at Fundeni Clinical Institute, Center of Uronephrology and Kidney Transplantation, between October 2018 and October 2019. Patients were followed for 12 months after KT. Exclusion criteria were: age <18 years, preformed AT1R-Ab, and follow-up period less than 1 year. In the aforementioned period, 67 patients were transplanted, but 7 patients were excluded because of preformed AT1R-Ab and 4 for graft failure before the end of the follow-up period, thus, 56 patients were included in the analysis (Figure 1). All patients included in the study were Caucasians. The study was approved by the Ethical Committee of Fundeni Clinical Institute (No. 41396/1 October 2018) and was conducted in accordance with the Helsinki Declaration of 1975.

### 2.2. Data Collection, Variables and Definition

Clinical, biological, and immunological data collected at the time of KT and during follow-up were divided in 5 major categories: recipient-associated, donor-associated, transplant-associated, treatment, and complications. Tacrolimus (TAC) trough concentrations at 2 weeks, 1, 3, 6, 9 and 12 months were evaluated. TAC level was reported as absolute value at different time-points, as mean concentration calculated based on all measurements within the 3–12 months after KT for every patient and as TAC intra-patient variability (IPV). TAC IPV was evaluated by calculating the coefficient of variation (CV) according to the formula: CV (%) = (standard deviation/mean TAC concentration within 6–12 months) × 100. Considering that the early phase after KT is associated with a wide fluctuation in TAC exposure, only TAC trough concentration measurements after 3 months were considered for mean TAC trough level calculation and after 6 months for IPV estimation. Three TAC trough concentration for each patient were assessed to estimate TAC IPV. High TAC IPV was defined as a CV value >30% [18]. Delayed graft function (DGF) was defined as graft disfunction that occurs in the first week post-KT and necessitates hemodialysis. Graft failure was defined as the absence of kidney function, occurring any time after transplantation due to irreversible graft injury requiring chronic dialysis and/or re-transplantation. Graft function was reported as estimated glomerular filtration rate (eGFR), evaluated with Modification of Diet in Renal Disease.

### 2.3. AT1R-Ab and HLA-Donor Specific Antibodies (HLA-DSA) Detection

Patients were screened for AT1R-Ab at the moment of KT and at 1 year after KT. The collected sera were stored at −80 °C until the day of measurement. AT1R-Ab detection was based on a quantitative ELISA technique (Celltrend, Luckenwalde, Germany), according to the manufacturer’s instructions. A threshold of >10 U/mL was considered for positivity based on the calibration curve, result from previous studies associated with negative graft outcomes, our previous results [17] and because 10 was the value of the third quartile. Microtiter 96-well polystyrene plates were coated with AT1R and in line with the manufacturer’s instructions, 100 μL of diluted samples were incubated at 2–8 °C for 2 h. After washing, the plates were incubated for 60 min with 100 μL of horse radish peroxidase-labelled anti-human IgG. After incubation with 100 μL of tetramethylbenzidine substrate for 20 min, optical absorbance of each well was measured at 450 nm by using an ELISA microplate reader.

The presence of HLA-DSA against HLA-A, -B, -C, -DR, -DP, -DQ was tested with the Luminex^®^ technique (One Lambda Inc., Canoga Park, CA, USA), prior to the KT as well as following KT, according to center protocol.

### 2.4. Immunosuppression Protocol

IS was conducted according to the center protocol. All patients received induction with anti-interleukin-2 receptor monoclonal antibody (anti-IL-2RmAb), Basiliximab 20 mg on day 0 and day 4, or with anti-thymocyte globulin (ATG) 1.5 mg/kg/day for 4 days. Maintenance IS consisted of triple therapy based on immediate-release tacrolimus (IR-TAC) or extended-release tacrolimus (ER-TAC), mycophenolate sodium (MPS), or mycophenolate mofetil (MMF) and glucocorticoids. The initial TAC dose was 0.2 mg/kg, administrated orally once daily (ER-TAC) or twice daily (IR-TAC). The subsequent doses were adjusted to achieve a TAC trough concentration between 8–10 ng/mL in the first 6 months and between 6–8 ng/mL until 1 year after KT. MPS/MMF initial dose was 2000 mg/1440 mg with subsequent reduction thereafter. Glucocorticoids consisted of intravenous methylprednisolone 500 mg on day 0 and day 1, 250 mg on day 2, 125 mg on day 3, then switched to 20 mg/day of oral prednisone up to the first month after KT and gradually tapered to 5 mg/day.

### 2.5. Statistical Analysis

Data were expressed as frequency and percentages for categorical variables, mean with standard deviation (SD), for continuous normally distributed variables and median with interquartile range (IQR) for continuous nonparametric variables. Chi-square or Fisher exact test were used as appropriate for categorical data comparison, *t* student test and Mann–Whitney U for continuous parametric and non-parametric data, respectively. Logistic regression analysis was performed to evaluate risk factors associated with de novo AT1R-Ab development at 1 year after KT. In the univariate model were included all variable with a *p*-value < 0.10 at group comparison. In the multivariate model, backward stepwise elimination method was used. ROC analysis was used to find the cut-off for mean TAC trough level as a risk factor. A *p*-value < 0.05 was considered statistically significant. The statistical analysis was performed with SPSS version 26 (SPSS Inc., Chicago, IL, USA).

## 3. Results

### 3.1. Patients’ General Characteristics

General characteristics of the patients are shown in Table 1. Among the 56 KT recipients mean age was 40.9 ± 10.4 years, 60.7% were males and the mean body mass index (BMI) was 22.8 ± 2.8 kg/m^2^. Nearly all patients had hypertension at the moment of KT (92.9%), 7.1% had type 2 diabetes, 12.5% had history of hepatitis B virus infection and 8.9% of hepatitis C virus infection. A previous KT was observed in 8.9% of cases and only 16.1% performed preemptive KT. The most frequent cause of chronic kidney disease (CKD) was glomerular disease (26.8%). Mean eGFR at 1 year after KT was 56.1 mL/min (41.3–66.5). Mean age of the donors was 47.9 ± 15.4 years, half of them were males and 62.5% of grafts came from cadaveric donors. Regarding transplant characteristics, median cold ischemia time (CIT) was 11.1 h (1.9–16.1) and in 28.6% of cases ≥4 HLA mismatches (MM) were observed. Most patients received induction IS with anti-IL-2mAb (83.9%), maintenance IS with IR-TAC was used in 53.6% of cases and MPS was preferred over MMF (89.3% vs. 10.7%). Mean TAC trough level was 8.7 ± 1.2 ng/mL, and a mean TAC >10 ng/mL was found in 10.7% of cases. Median TAC IPV estimated with CV was 14% (5.5–22.6%) and a high TAC IPV (>30%) was observed in 8 patients (14.3%). During follow-up, two patients developed DGF, no biopsy-proven rejection (BPR) cases were reported, and four patients developed BK viremia.

### 3.2. Incidence and Characteristics of Patients with De Novo AT1R-Ab

Twelve out of fifty-six patients (21.4%) developed de novo AT1R-Ab at 1 year after KT, and the median value of these antibodies in the entire cohort was 8.5 U/mL (6.8–10.4). The median titer of AT1R-Ab in the positive group was 15.6 U/mL (10.8–19.8) (Figure 2). Characteristics of the patients with de novo AT1R-Ab are provided in Table 1. Compared to patients without de novo AT1R-Ab, those with de novo AT1R-Ab had a lower mean BMI (21.4 ± 1.8 vs. 23.3 ± 2.9 kg/m^2^, *p* = 0.06), a longer period on dialysis before KT (33 (4.5–90.7) vs. 15 months (1.3–30.7), *p* = 0.12)) and a higher percentage of ≥4 HLA-MM (50% vs. 22.7%, *p* = 0.08). Additionally, patients with de novo AT1R-Ab received significantly more often induction IS with ATG (41.7% vs. 9.1%, *p* = 0.01) and maintenance with IR-TAC (83.3% vs. 45.5%, *p* =0.02). Furthermore, patients from this group had a significantly higher mean TAC trough level throughout the first year after KT (9.5 ± 1.7 vs. 8.4 ± 0.9 ng/mL, *p* = 0.01), a significantly higher percent of mean TAC trough level >10 ng/mL (41.7% vs. 11.4%, *p* = 0.02) and higher median TAC IPV (19.7 (IQR:11.5–34.2) vs. 12.3% (IQR: 4.4–21.0), *p* = 0.05) and TAC IPV > 30% (33.3% vs. 9.1%, *p* = 0.05), but at the limit of significance. There was no difference in terms of recipient age, gender, comorbidities, causes of CKD, angiotensin receptor blocker (ARB) treatment, donor characteristics, ischemia times and BK viremia.

### 3.3. Risk Factors for De Novo AT1R-Ab Development

To assess the risk factors associated with de novo AT1R-Ab formation at 1 year after KT, logistic regression analysis was performed (Table 2). On univariate analysis, recipient BMI (OR = 0.74, 95% CI: 0.55–0.99, *p* = 0.04), ATG induction (OR = 7.14, 95% CI: 1.53–33.39, *p* = 0.01), IR-TAC (OR = 6.00, 95% CI: 1.17–30.62, *p* = 0.03), and mean TAC trough level (OR = 2.01, 95% CI: 1.10–3.66, *p* = 0.05) were significantly associated with de novo AT1R-Ab formation. The high TAC IPV (>30%) was at the limit of significance (OR = 5.00, 95% CI: 1.00–24.27, *p* = 0.05). On multivariate analysis, ATG induction therapy (OR = 7.20, 95% CI: 1.30–39.65, *p* = 0.02), IR-TAC (OR = 6.20, 95% CI: 1.16–41.51, *p* = 0.03) and mean TAC trough level (OR = 2.36, 95% CI: 1.14–4.85, *p* = 0.01) were identified as independent risk factors for de novo AT1R-Ab formation. Using a ROC analysis, we found that a mean TAC trough level >10 ng/mL had an area under the curve of 0.70 (95% CI: 0.48–0.89, *p* = 0.04) as a risk factor.

## 4. Discussion

In the current prospective study, we sought to evaluate the incidence of de novo AT1R-Ab and the potential risk factors associated with their development, at 1 year after KT. We observed that de novo AT1R-Ab incidence in our cohort was 21.4%, which falls within the range reported in previous studies [5]. Lefaucheur et al., showed in the largest prospective study that evaluated AT1R-Ab, using a cut-off value for antibody detection of >10 U/mL, that post-transplant AT1R-Ab incidence was 27.3% at 1 year after transplantation [16]. Pinelli et al. reported a closer incidence (18.8%) to our results, in a prospective study of 142 KT recipient from living donors [19]. One retrospective study performed in pediatric patients showed a de novo AT1R-Ab prevalence of 26% [20]. Lower frequencies of de novo AT1R-Ab were reported in the studies of Taniguchi et al. (3%), and Gareau et al. (16%) [10,12]. Distinct to our study, all the above-mentioned studies included patients with different percentages of HLA-DSA, either preformed or de novo. In our cohort, none of the patients had preformed HLA-DSA, and only one patient, from the AT1R-Ab negative group, developed de novo HLA-DSA.

The main finding of our study was related to the type of induction and maintenance IS as significant determinants for de novo AT1R-Ab development. Previous studies showed that age, male gender, number of previous KT, diabetes, cause of CKD produced by lupus nephritis or focal segmental glomerular sclerosis and living donation were factors associated with preformed AT1R-Ab [13,21,22]. Compared to the preformed type, in the formation of de novo AT1R-Ab may be involved specific transplant-associated factors. Endothelial injury and neoantigen AT1R exposure, appropriate conditions for autoimmunity through loss of self-tolerance, and the interplay between autoimmunity and alloimmunity are among the key elements involved in the complex mechanism of AT1R-Ab formation [1,2,8]. Some transplant-associated conditions could be responsible for endothelial damage, AT1R exposure as a cryptic antigen, and initiation of AT1R-Ab formation. Inflammatory conditions such as ischemia-reperfusion injury (IRI), infections (e.g., BK nephropathy), presence of HLA-DSA, rejection and IS may be factors for endothelial injury and could trigger de novo AT1R-Ab formation. IRI is a process present inevitably in all KT patients, responsible for both the occurrence of immunogenicity and cellular lesions of prognostic importance for graft survival. Prolonged CIT as a marker of IRI and endothelial damage has been described as an independent predictor for HLA antibodies formation and could also be proposed as a determinant for de novo AT1R-Ab development [1,23]. Even though we observed a higher CIT in AT1R-Ab group, this was not statistically significant and was not found as a risk factor. Regarding AT1R-Ab and infections, initially, parvovirus infection was speculated to be involved in antibody formation by antigen mimicry and cross-reactivity, but this was later disproven [24]. BK nephropathy is an infectious disease of the graft associated with inflammation of the tubulointerstitial compartment and graft damage [25]. Endothelial lesions of the glomerular and peritubular capillaries associated with the presence of BK nephropathy have also been described [26,27]. We found four patients with BK viremia during follow-up, only one being in the AT1R-Ab positive group. The number of events was too low to demonstrate any difference. Similarly, Pear et al. reported no difference between patients with or without de novo AT1R-Ab regarding BK viremia [20]. Inflammation associated with HLA-DSA, or rejection could lead to graft injury, increased expression on AT1R neoantigens and AT1R-Ab formation. The relationship between AT1R-Ab and HLA-DSA seems to be mutual regarding their development and points out the interaction between allo- and autoimmunity [5,8]. This causal relationship could not be confirmed in our study, because none of the patients had preformed HLA-DSA and only one patient developed de novo HLA-DSA, who was part of the de novo AT1R-Ab negative group. Additionally, the relationship with rejection has not been proven, because we did not find any case of BPR. These findings are similar to those found by Crespo et al., who showed no association between post-transplant AT1R-Ab and HLA-DSA detection or biopsy-proven anti-body-mediated rejection [28].

Data regarding the role of IS on the development of de novo AT1R-Ab are lacking.

We found that use of ATG increased the risk of AT1R-Ab development 7.2 times. This is consistent with the finding of Pearl et al., in a pediatric retrospective cohort which showed that AT1R-Ab presence in the first 2 years after KT was associated with ATG induction regimen, but the association was limited to a comparison analysis, since this was not the main purpose of the study [20]. From a mechanistic perspective, this seems to be paradoxical at first sight, given the ATG immunosuppressive effects. By acting primarily on T-cell depletion from the blood and peripheral lymphoid tissues, including T helper cells, by promoting B-cell apoptosis and due to the immunomodulatory effects on the endothelium, ATG protects against endothelial damage and autoantibody formation [29]. Nonetheless, a possible argumentation for the causative link between ATG and AT1R-Ab may result from the complete and prolonged depletion of the regulatory T cell subset (CD4^+^ CD25^+^), thus leading to the loss of self-tolerance and autoimmunity [30,31].

We observed that IR-TAC and increased mean TAC trough level between 3–12 months after KT were independent risk factors for de novo AT1R-Ab formation. According to our knowledge, these observations were reported for the first time. TAC has a narrow therapeutic index, high IPV and fluctuations in trough levels [32]. Among TAC adverse events, nephrotoxicity in an important issue, characterized by tubulointerstitial lesions, endothelial injury, disfunction, vascular damage and decreased graft function, which may appear even in patients with trough levels within the therapeutic range [33]. A recent systematic review and meta-analysis, including 11 randomized-control trials, showed no difference between IR-TAC and ER-TAC regarding the impact on graft function at 1 year after KT [34]. In our study there was no difference in terms of eGFR at 1 year, according to AT1R-Ab status. When we analyzed eGFR according to the TAC type, we found that patients treated with IR-TAC had a significantly more decreased eGFR than those treated with ER-TAC at 1 year (50.0 (39.9–62.3) vs. 60.1 mL/min (48.6–71.3), *p* = 0.03)). Additionally, we analyzed the eGFR at 1 year according to AT1R-Ab status and TAC type and we observed that in patients without AT1R-Ab there was no difference between IR-TAC and ER-TAC in terms of eGFR (52.4 (38.7–41.8) vs. 63.7 mL/min (48.6–77.7), *p* = 0.159) and in patients with AT1R-Ab eGFR was significantly lower in patients treated with IR-TAC (40.0 (38.7–41.8) vs. 63.7 mL/min (48.6–77.7), *p* = 0.04), but the latter result should be interpreted with caution given the small number of patients (2 vs. 10 patients). Mean TAC trough concentration, but not IPV, was significantly higher in patients with AT1R-Ab. However, median TAC IPV and IPV >30% were higher in patients with AT1R-Ab and at the limit of significance (*p* = 0.05). No significant difference regarding mean TAC trough level (AT1R-Ab positive group, IR-TAC: 10.2 ± 0.1 ng/mL vs. ER-TAC: 9.3 ± 1.9 ng/mL, *p* = 0.56; AT1R-Ab negative group, IR-TAC: 8.5 ± 1.1 ng/mL vs. 8.4 ± 0.7 ng/mL, *p* = 0.88) and median TAC IPV (AT1R-Ab positive group, IR-TAC: 19.7% (12.2–35.5) vs. ER-TAC: 15.2% (3.4–15.2%), *p* = 0.48; AT1R-Ab negative group, IR-TAC: 13.9% (7.1–21) vs. ER-TAC: 12.1 (3–22.7), *p* = 0.63)) according to AT1R-Ab and TAC type has been observed. Based on this observation, we supposed that IR-TAC may be involved in de novo AT1R-Ab formation through their nephrotoxic action, especially on the endothelium, a statement that could be sustained by the lower eGFR in this subgroup of patients, but not confirmed in the absence of graft biopsy.

Taken together, our results provide new insights into the field of non-HLA antibodies, through the role of IS in the development of de novo AT1R-Ab. Induction IS with ATG and maintenance IS with IR-TAC, especially if the latter reaches high concentrations, could create a possible scenario which favors loss of self-tolerance, neoantigens formation and AT1R-Ab development. Certainly, these observations need to be confirmed in larger prospective studies and RCTs.

Our study has some limitations. First, this was an observational, non-interventional study, thus medication was administered according to attending physician approach which may limit the generatability of the results. Second, another limitation is represented by the small sample size, which demands caution in interpreting the results. Third, inferences arise from a single-center study. Fourth, the study period was limited to 1 year of follow-up. Fifth, the approach of the center to perform indication biopsies led to no cases of biopsy-proven rejection. The strengths were the prospective design of the study, multivariate analysis, and mean TAC trough level and IPV variable used to characterize the role of TAC as a risk factor.

## 5. Conclusions

In conclusion, our prospective study showed an incidence of 21.4% for de novo AT1R-Ab and that induction immunosuppression with ATG, maintenance immunosuppression with IR-TAC and the TAC trough level are independent risk factors for antibody development in KT recipients. These observations may lead to consideration of the immunosuppression type in the evaluation of immunological risk for AT1R-Ab development, careful monitoring of TAC concentrations, and consideration of the use of ER-TAC in patients in whom induction has been made with ATG or the conversion from IR-TAC to ER-TAC in the first year after KT. Further studies should address these assumptions.

## Figures and Tables

**Figure 1 jcm-10-05390-f001:**
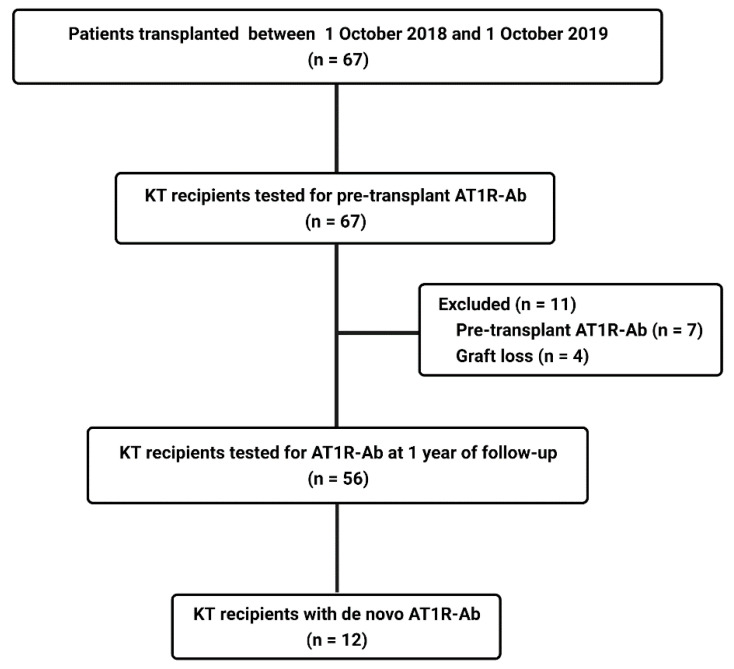
Study flow diagram.

**Figure 2 jcm-10-05390-f002:**
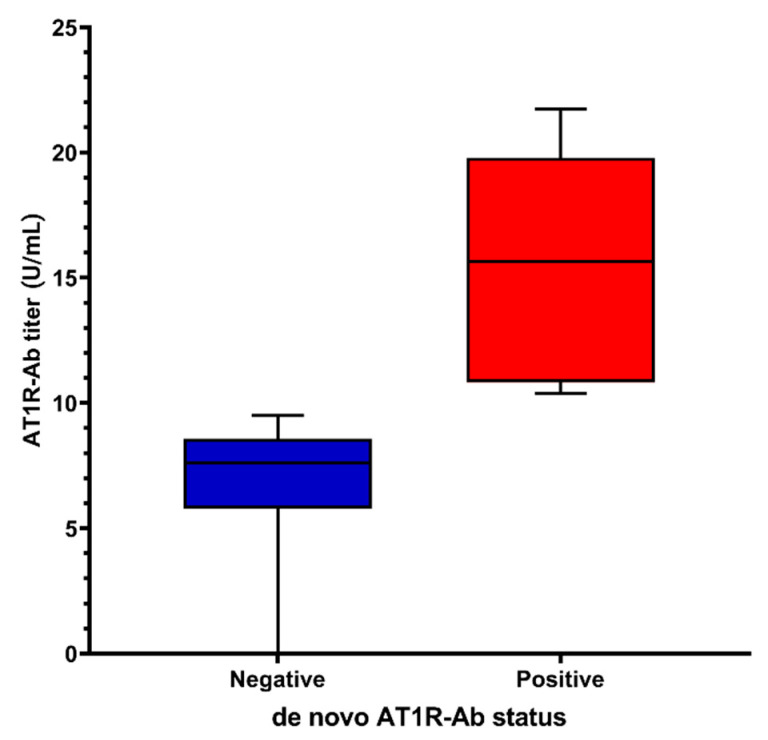
Antibody titers in positive and negative de novo AT1R-Ab groups.

**Table 1 jcm-10-05390-t001:** Cohort characteristics and those according to the presence of de novo AT1R-Ab.

Variables	Entire Cohort (n = 56)	(−) Novo AT1R-Ab (n = 44)	(+) De Novo AT1R-Ab (n = 12)	*p*-Value
**Recipient characteristics**				
Age (mean, years)	40.9 ± 10.4	40.8 ± 10.3	41.5 ± 11.3	0.84
Gender (%)				0.50
Male	34 (60.7%)	28 (63.6%)	6 (50%)	
Female	22 (39.3%)	16 (36.4%)	6 (50%)	
Body mass index (mean, kg/m^2^)	22.8 ± 2.8	23.3 ± 2.9	21.4 ± 1.8	0.06
**Comorbidities**				
Hypertension (%)	52 (92.9%)	40 (90.9%)	12 (100%)	0.56
Diabetes (%)	4 (7.1%)	1 (4.5%)	2 (16.7%)	0.20
Active/past smoker (%)	15 (26.8%)	13 (29.5%)	2 (16.7%)	0.48
Hepatitis B virus infection (%)	7 (12.5%)	6 (13.6%)	1 (8.3%)	1
Hepatitis C virus infection (%)	5 (8.9%)	4 (9.1%)	1 (8.3%)	1
History of tuberculosis (%)	4 (7.1%)	2 (4.5%)	2 (16.7%)	0.20
History of allergy (%)	11 (19.6%)	9 (20.5%)	2 (16.7%)	0.77
History of blood transfusion (%)	23 (41.4%)	18 (40.9%)	5 (41.7%)	1
Previous KT (%)	5 (8.9%)	3 (6.8%)	2 (16.7%)	0.29
History of pregnancy *	12/22 (54.5%)	8/16 (50%)	4/6 (66.7%)	0.64
Hemodialysis (%)	45 (80.4%)	34 (77.3%)	11 (91.7%)	0.42
Peritoneal dialysis (%)	2 (3.6%)	1 (2.7%)	1 (8.3%)	0.42
Dialysis period (median, months)	15.5 (2–39)	15 (1.3–30.7)	33 (4.5–90.7)	0.12
Preemptive KT (%)	9 (16.1%)	8 (18.2%)	1 (8.3%)	0.66
Chronic kidney disease cause (%)			0.38	
Glomerular disease	15 (26.8%)	9 (20.5%)	6 (50%)	
Diabetes kidney disease	2 (3.6%)	1 (2.3%)	1 (8.3%)	
Autosomal dominant polycystic kidney disease	8 (14.3%)	7 (15.9%)	1 (8.3%)	
Tubulointerstitial disease	10 (17.9%)	9 (20.4%)	1 (8.3%)	
Others	2 (3.6%)	2 (4.5%)	0 (%)	
Unknown	19 (33.9%)	16 (36.4%)	3 (25%)	
**Graft function**				
Baseline creatinine (median, mg/dL)	3.7 (1.8–5.7)	3.5 (1.9–5.7)	4.4 (1.3–6.2)	0.72
Baseline eGFR (median, mL/min/1.73 m^2^)	20.7 (10.8–38.2)	21 (10.6–38.2)	15.7 (10.8–48.3)	0.75
1-year creatinine (median, mg/dL)	1.3 (1.1–1.7)	1.3 (1.2–1.8)	1.3 (0.9–1.6)	0.33
1-year eGFR (mL/min/1.73 m^2^)	56.1 (41.3–66.5)	54.4 (41.3–63.5)	60.6 (42.4–77.1)	0.42
**Donor characteristics**				
Age (mean, years)	47.9 ± 15.4	47.8 ± 14.7	48.6 ± 18.4	0.86
Gender (%)				0.74
Male	28 (50%)	21 (47.7%)	7 (58.3%)	
Female	28 (50%)	23 (52.3%)	5 (41.7%)	
Donor type (%)				1
Cadaveric	35 (62.5%)	27 (61.4%)	8 (66.7%)	
Living	21 (37.5%)	17 (38.6%)	4 (33.3%)	
**Transplant characteristics**				
Cold ischemia time (median, hours)	11.1 (1.9–16.1)	10.7 (1.9–14.2)	11.7 (2.3–17.4)	0.43
Warm ischemia time (mean, mins)	28.6 ± 3.8	28.6 ± 4.0	28.6 ± 3.2	0.95
HLA-MM ≥ 4 (%)	16 (28.6%)	10 (22.7%)	6 (50%)	0.08
Preformed HLA-DSA (%)	0 (0%)	0 (0%)	0 (0%)	-
De novo HLA-DSA (%)	1 (1.8%)	1 (2.3%)	0 (0%)	1
**Treatment characteristics**				
Induction therapy (%)				**0.01**
Anti-thymocyte globulin	9 (16.1%)	4 (9.1%)	5 (41.7%)	
Anti-IL2RmAb	47 (83.9%)	40 (90.9%)	7 (58.3%)	
Maintenance therapy				**0.02**
Calcineurin inhibitors (%)				
Immediate release TAC	30 (53.6%)	20 (45.5%)	10 (83.3%)	
Extended-release TAC	26 (46.4%)	24 (54.5%)	2 (16.7%)	
Antimetabolite (%)				0.59
Mycophenolate sodium	50 (89.3%)	40 (90.9%)	10 (83.3%)	
Mycophenolate mofetil	6 (10.7%)	4 (9.1%)	2 (16.7%)	
TAC trough level (C0)				
3 months (mean, ng/mL)	12.0 ± 3.8	11.5 ± 3.1	13.8 ± 5.6	0.05
6 months (mean, ng/mL)	8.1 ± 1.7	7.9 ± 1.6	8.5 ± 2.1	0.32
9 months (mean, ng/mL)	7.5 ± 1.3	7.3 ± 1.2	8.0 ± 1.6	0.13
12 months (mean, ng/mL)	7.2 ± 1.6	7.1 ± 1.5	7.6 ± 2.1	0.39
Mean TAC trough level (ng/mL)	8.7 ± 1.2	8.4 ± 0.9	9.5 ± 1.7	**0.01**
Mean TAC > 10 ng/mL (%)	10 (17.9%)	5 (11.4%)	5 (41.7%)	**0.02**
Median TAC IPV, CV%	14.0 (5.5–22.6)	12.3 (4.5–21.0)	19.7 (11.5–34.2)	0.05
Median TAC IPV > 30%	8 (14.3%)	4 (9.1%)	4 (33.3%)	0.05
Angiotensin II receptor blocker (%)	3 (5.3%)	2 (4.5%)	1 (8.3%)	1
**Complications**				
Delayed graft function (%)	2 (3.6%)	2 (4.5%)	0 (0%)	1
Rejection	0 (0%)	0 (0%)	0 (0%)	-
BK viremia	4 (7.1%)	3 (6.8%)	1 (8.3%)	1

n—number; KT—kidney transplant; eGFR—estimated glomerular filtration rate; HLA-MM—human leucocyte antigen mismatch; HLA-DSA—specific antibodies against human leucocyte antigen; Anti-IL-2RmAb—interleukin 2 receptor; TAC—tacrolimus; IPV—intra-patient variability; (−)—absent; (+)—present; * total number of female patients was 22.

**Table 2 jcm-10-05390-t002:** Logistic regression analysis to evaluate risk factor for de novo AT1R-Ab development.

	Univariate Logistic Regression			Multivariate Logistic Regression		
Variables	OR	95% CI	*p*-Value	OR	95% CI	*p*-Value
Recipient BMI	0.74	0.55–0.99	0.04	-	-	-
HLA-MM ≥ 4	3.40	0.89–12.89	0.07	-	-	-
Dialysis period	1.003	0.99–1.01	0.50	-	-	-
Anti-thymocyte globulin	7.14	1.53–33.39	0.01	7.20	1.30–39.65	0.02
Immediate-release TAC	6.00	1.17–30.62	0.03	6.20	1.16–41.51	0.03
Mean TAC trough level	2.01	1.10–3.66	0.007	2.36	1.14–4.85	0.01
Median TAC IPV >30%	5.00	1.03–24.27	0.05	-	-	-

OR—odds ratio; CI—confidence interval; BMI—body mass index; HLA-MM—human leukocyte antigen mismatch; TAC—tacrolimus; IPV—intra-patient variability.

## Data Availability

The datasets generated and/or analyzed during the current study are available from the corresponding author on reasonable request.

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
