# Peer review of "Immunosuppression as a Risk Factor for De Novo Angiotensin II Type Receptor Antibodies Development after Kidney Transplantation"

_jcm, 2021, doi:10.3390/jcm10225390_

Round 1
Reviewer 1 Report
Thank you for the chance to review this interesting article.
This study revealed the risk factor for the angiotensin 2 type receptor antibodies.
However, there was a serious problem in the statistical method.
The number of event, angiotensin 2 type receptor antibodies production in this study, was only 12 cases. In such condition, it is impossible to perform multivariate analysis. And the conclusion might be incorrect.
Author Response
Reviewer #1
Thank you for the chance to review this interesting article. This study revealed the risk factor for the angiotensin 2 type receptor antibodies. However, there was a serious problem in the statistical method. The number of events, angiotensin 2 type receptor antibodies production in this study, was only 12 cases. In such condition, it is impossible to perform multivariate analysis. And the conclusion might be incorrect.
We are thankful for your comments. We agree that the sample size was small and this fact was mentioned as the main limitation of the study. Also, we mentioned that the results should be interpreted in this context. Nevertheless, at this moment there is no guideline or consensus regarding the perfect number of a sample size at which a multivariate regression analysis should be performed. Additionally, there are previous studies that evaluated AT1R-Ab with similar sample sizes in which multivariate analysis was performed (PMID: 29444269; PMID: 29451342). Moreover, increasing the sample size does not necessarily guarantee an increase in the number of AT1R-Ab cases.
Reviewer 2 Report
In this article, the authors address a topic of interest: the development of non-HLA antibodies (specifically AT1R-Ab) and their risk factors. In their cohort of 56 recipients, 21.4% develop AT1R-Ab. Immunosuppression with anti-thymocyte globulin, maintenance immunosuppression with immediate-release tacrolimus, and mean tacrolimus trough levels are independent risk factors for de novo AT1R-Ab at one year after kidney transplant in their cohort. Although the question evaluated is doubtless a subject of interest to the transplant community that deserves more studies, some major concerns should be addressed.
MAJOR COMMENTS
- Some immunological aspects such as peak PRA or sensitizing events have not been reported. These factors could explain the increased use of anti-thymocyte globulin induction immunosuppression in patients who develop de novo AT1R-Ab. Therefore, they should be assessed in the logistic regression analysis.
- The authors should explain which criteria have been considered when using immediate-release TAC vs. extended-release TAC. Moreover, TAC trough concentration for each tacrolimus group should be specified. They could hypothesize if IR-TAC were involved in the development of de novo AT1R-Ab because of a higher TAC trough concentration in this group or another explanation.
- In the discussion section, the authors assess possible argumentation regarding the causative relation between ATG and AT1R. In a previous review (PMID 33002671), the authors postulated that “Depleting agents could decrease the titer and de novo AT1R-Ab formation after KT by targeting B cells and plasma cells and possibly by endothelial immunomodulation […]”. They should discuss these statements in the present study.
- The study sample is too small, and only 12 patients developed AT1R-Ab. Therefore, these results should be taken with caution due to their limitations. For example, the relationship between HLA-DSA and ATR-Ab could not be assessed. Please consider increasing the study cohort.
MINOR COMMENTS
- Materials and Methods, section 2.3. Please review the sentence “All patients were screened for AT1R-Ab at the moment of KT and those who reached 1 year of follow-up for de novo AT1R-Ab.” It was previously specified in section 2.1. that a follow-up period lower than 1 year was an exclusion criterion.
- Results, section 3.1. and Table 1. Please consider expressing cold ischemia time in hours instead of minutes.
- Results, section 3.3. “[…] an aria under the curve of 0.70 […]” should be “[…] an area under the curve of 0.70 […]”.
- Some recent missing references discuss the relationship between de novo AT1R-Ab and HLA-DSA or ABMR histology: PMID: 34305943
Author Response
Reviewer #2
In this article, the authors address a topic of interest: the development of non-HLA antibodies (specifically AT1R-Ab) and their risk factors. In their cohort of 56 recipients, 21.4% develop AT1R-Ab. Immunosuppression with anti-thymocyte globulin, maintenance immunosuppression with immediate-release tacrolimus, and mean tacrolimus trough levels are independent risk factors for de novo AT1R-Ab at one year after kidney transplant in their cohort. Although the question evaluated is doubtless a subject of interest to the transplant community that deserves more studies, some major concerns should be addressed.
MAJOR COMMENTS
Some immunological aspects such as peak PRA or sensitizing events have not been reported. These factors could explain the increased use of anti-thymocyte globulin induction immunosuppression in patients who develop de novo AT1R-Ab. Therefore, they should be assessed in the logistic regression analysis.
We would like to thank you for the constructive comments and suggestions. Sensitizing events such as previous kidney transplantation (KT), history of blood transfusions and history of allergy were mentioned in Table 1. After your suggestion we added in Table 1 data regarding other sensitizing events such as history of pregnancy and HLA-DSA.
ATG was used for patients with a previous KT, in those who received a graft from a cadaveric donor with prolonged CIT and for those with IgA nephropathy with extracapillary proliferation on the native kidneys. Because no significant difference (or even near) has been observed regarding these sensitizing factors, between patients with and without de novo AT1R-Ab, these variables were not included in the regression analysis. Only variables with p value < 0.10 at comparison analysis were included in the regression model.
We do not have data regarding PRA. HLA-DSA are identified based on Luminex technique.
The authors should explain which criteria have been considered when using immediate-release TAC vs. extended-release TAC. Moreover, TAC trough concentration for each tacrolimus group should be specified. They could hypothesize if IR-TAC were involved in the development of de novo AT1R-Ab because of a higher TAC trough concentration in this group or another explanation.
As we mentioned in the discussion section, this was an observational, non-interventional study, thus medication was administered according to attending physician approach, which may limit the generalizability of the results. As a non-observational study, no criteria for medication allocation could be considered. The approach in our Center is to use IR-TAC as a first choice, and when IR-TAC is unavailable or some issues regarding compliance are present, ER-TAC is used instead of IR-TAC.
Regarding the TAC trough concentration for each TAC type group according to AT1R-Ab status, we added at the discussion section the following statement: “Mean TAC trough concentration, but not IPV, was significantly higher in patients with AT1R-Ab. However, median TAC IPV and IPV >30% were higher in patients with AT1R-Ab and at the limit of significance (p=0.05). No difference regarding mean TAC trough level and IPV according to AT1R-Ab and TAC type has been observed.” In this revised version we added statistical data regarding mean TAC and median TAC IPV for each TAC type according to de novo AT1R-Ab status, as per your request. (Rows: 279-283).
In the discussion section, the authors assess possible argumentation regarding the causative relation between ATG and AT1R. In a previous review (PMID 33002671), the authors postulated that “Depleting agents could decrease the titer and de novo AT1R-Ab formation after KT by targeting B cells and plasma cells and possibly by endothelial immunomodulation […]”. They should discuss these statements in the present study.
In the Discussion section, 3rd paragraph, we mentioned some aspects regarding the effect of ATG as a depleting and immunomodulatory agent, that were similar to those you mentioned, used in our previous review article: “From a mechanistic perspective, this seems to be paradoxical at first sight, given the ATG immunosuppressive effects. By acting primarily on T-cell depletion from the blood and peripheral lymphoid tissues, including T helper cells, by promoting B-cell apoptosis and due to the immunomodulatory effects on the endothelium, ATG protects against endothelial damage and autoantibody formation”. Afterwards we tried to offer a pertinent explanation for the current opposite scenario where ATG could be involved in Ab formation: “Nonetheless, a possible argumentation for the causative link between ATG and AT1R-Ab may result from the complete and prolonged depletion of the regulatory T cell subset (CD4+ CD25+), thus leading to the loss of self-tolerance and autoimmunity’’.
The study sample is too small, and only 12 patients developed AT1R-Ab. Therefore, these results should be taken with caution due to their limitations. For example, the relationship between HLA-DSA and ATR-Ab could not be assessed. Please consider increasing the study cohort.
We agree with the small sample size comment and we mentioned it in the limitations paragraph, from the discussions section. Also, in this revised version, we added the following statement: “Second, another limitation is represented by the small sample size, which demands caution in interpreting the results.’’ (Rows 297-298)
Regarding the relationship between HLA-DSA and AT1R-Ab, there is no limitation. The detection of preformed HLA-DSA was performed, but no case was found pre-transplantation. Also, patients were monitored regularly (at every 3 months post-KT) for de novo HLA-DSA development and only one patient had de novo HLA-DSA, which belonged to the AT1R-Ab negative group. In this case, the relationship has been evaluated and no association has been observed.
For future studies on the AT1R-Ab subject we will take into consideration the expansion of the sample size.
MINOR COMMENTS
Materials and Methods, section 2.3. Please review the sentence “All patients were screened for AT1R-Ab at the moment of KT and those who reached 1 year of follow-up for de novo AT1R-Ab.” It was previously specified in section 2.1. that a follow-up period lower than 1 year was an exclusion criterion.
The sentence was modified as follows:“Patients were screened for AT1R-Ab at the moment of KT and at 1 year after KT” (Material and methods, section 2.3., Row: 93)
Results, section 3.1. and Table 1. Please consider expressing cold ischemia time in hours instead of minutes.
CIT expression was modified accordingly in Table 1 and at the results section (3.1. subsection, Row 147).
Results, section 3.3. “[…] an aria under the curve of 0.70 […]” should be “[…] an area under the curve of 0.70 […]”.
The word “aria” was corrected to “area”.
Some recent missing references discuss the relationship between de novo AT1R-Ab and HLA-DSA or ABMR histology: PMID: 34305943
Thank you for this interesting and recent study suggestion. Indeed, in this study, Crespo et al. showed an interesting relationship between that AT1R-Ab, HLA-DSA and ABMRh. They showed that preformed AT1R-Ab are strongly associated with HLA-DSA (preformed or de novo) and with ABMRh DSApos. On the other hand, post-transplant AT1R-Ab were not associated with HLA-DSA detection or with ABMRh. The latter information was used to comment the relatively similar finding from our study. The following statement was added to the discussion section: “These findings are similar to those found by Crespo et al. who showed no association between post-transplant AT1R-Ab and HLA-DSA detection or biopsy-proven anti-body-mediated rejection” (rows 241-244).
Reviewer 3 Report
Sorohan and colleagues conduct the current study “Immunosuppression as a risk factor for de novo angiotensin II type receptor antibodies development after kidney transplantation” which demonstrated the induction IS regimens and TAC levels may act as risk factors for de novo AT1R-Ab. This study has been proposed that the kidney injury from high TAC levels may cause to endothelial injury and develop de novo AT1R-Ab. Although this is an interesting new point hypothesis, however, some concerns should be addressed. Here are some major concerns.
- It was unclear that when was the timing of AT1R-Ab were assessed because the authors did describe only a time point. Indeed, I couldn’t find the values of positive cases. Thus, I suggest demonstrating as a bar graph +/- individual values rather than descriptive results.
- Since the cut-off point of positive AT1R-Ab in this study was 10 U/mL, it is hardly convincing that the median values of AT1R-Ab 8.5 U/ml (6.8-10.4) from entire cohort (n= 56) were also included the positive case of AT1R-Ab (12 recipients). Again, more data clarification is needed. The authors have to explicit the AT1R-Ab values in positive case as well.
- Do the mean TAC>10 ng/ml (table 1) stands for the mean from over 12-month duration or from how many months?
- KT recipients’ blood pressure is one of the most influence factor for de novo AT1R-Ab. The authors should do clarify.
- How can the authors explain the ATG showed significant association with de novo AT1R-Ab? Is it related to second KT or the allograft quality? Is it possible explained that because the recipients have a high immunological risk, therefore, they received ATG and maintained with higher IS levels in the first year post-KT? If the authors have not proved for the true causal effects, they should tone down or made change wordings in the title as well as in the discussion part.
- When was the time of baseline GFR in Table 1? Normally, the baseline GFR should be addressed at the time of steady state.
- Currently, the HLA-DSA seems to be an established risk factor for graft deterioration, thus, the authors should mention as limitation of this study.
- The authors should provide some evidence from their study to support why the ER-TAC regimen showed more decreased eGFR compared with IR-TAC. For example, trough TAC levels, TAC IPV between groups, etc.
- There are some typing errors, e.g, area (L183)
- Please do recheck and complete Ref. #33.
Author Response
Reviewer #3
Sorohan and colleagues conduct the current study “Immunosuppression as a risk factor for de novo angiotensin II type receptor antibodies development after kidney transplantation” which demonstrated the induction IS regimens and TAC levels may act as risk factors for de novo AT1R-Ab. This study has been proposed that the kidney injury from high TAC levels may cause to endothelial injury and develop de novo AT1R-Ab. Although this is an interesting new point hypothesis, however, some concerns should be addressed. Here are some major concerns.
1. It was unclear that when was the timing of AT1R-Ab were assessed because the authors did describe only a time point. Indeed, I couldn’t find the values of positive cases. Thus, I suggest demonstrating as a bar graph +/- individual values rather than descriptive results.
We really appreciate your constructive comments and suggestions. AT1R-Ab were first assessed pre-transplant to find preformed AT1R-Ab cases, which were excluded, and were also tested at 1 year after transplantation (each patient), to identify the presence of de novo AT1R-Ab. A bar graph was made to illustrate AT1R-Ab titers for negative and positive groups (Figure 1, results section, subsection 3.2).
2. Since the cut-off point of positive AT1R-Ab in this study was 10 U/mL, it is hardly convincing that the median values of AT1R-Ab 8.5 U/ml (6.8-10.4) from entire cohort (n= 56) were also included the positive case of AT1R-Ab (12 recipients). Again, more data clarification is needed. The authors have to explicit the AT1R-Ab values in positive case as well.
Values into the brackets represented IQR and not min-max. So, 10.4U/ml represented the value of the third quartile. To clarify, a bar graph was made to show values for both groups (AT1R-Ab positive vs negative). The titer for the AT1R-Ab negative group was 7.5 U/ml (5.7- 8.4) and for the AT1R-Ab positive group was 15.6 U/ml (10.8- 19.8). Data regarding AT1R-Ab titer in the positive group were added to the results section and to the abstract.
3. Do the mean TAC>10 ng/ml (table 1) stands for the mean from over 12-month duration or from how many months?
“Mean TAC > 10 ng/ml” variable stand for the mean TAC levels measured between months 3-12 (the same as for the continuous variable “mean TAC trough level”). Considering that the early phase after KT is associated with a wide fluctuation in TAC exposure, only TAC concentrations measurement after 3 months were considered for mean TAC trough level and mean TAC> 10 ng/ml calculation (mentioned in the Material and method section, 2.2. Data collection, variables and definition).
4. KT recipients’ blood pressure is one of the most influence factor for de novo AT1R-Ab. The authors should do clarify.
Hypertension (HTN) has been described as a clinical manifestation of rejection mediated by AT1R-Ab, but not as a risk factor for Ab formation, and, moreover, this finding was inconstant across the studies. In the seminal paper of Dragun et al., published in 2005, vascular rejection mediated by AT1R-Ab was associated with a clinical phenotype of severe HTN. HTN was also described in a few other papers published thereafter as a manifestation of rejection mediated by AT1R-Ab, but not confirmed in other studies. Anyway, when present, HTN was always associated with rejections, but not with the presence of AT1R-Ab alone. In our cohort, almost all patients were hypertensive (92.9%) at the moment of KT and implicitly before de novo AT1R-Ab formation and none had biopsy-proven rejection. The prevalence of HTN was approximately the same between patients that developed and those who did not develop de novo AT1R-Ab (100% vs 92.9%). Based on the above explanations, we considered that HTN could not be a risk factor for AT1R-Ab formation.
5. How can the authors explain the ATG showed significant association with de novo AT1R-Ab? Is it related to second KT or the allograft quality? Is it possible explained that because the recipients have a high immunological risk, therefore, they received ATG and maintained with higher IS levels in the first year post-KT? If the authors have not proved for the true causal effects, they should tone down or made change wordings in the title as well as in the discussion part.
A mechanistic explanation regarding the association between ATG and de novo AT1R-Ab formation was provided at the discussion section. Regarding ATG use, it was indeed administered to patients with a previous KT, DD with prolonged CIT and IgA nephropathy with extracapillary proliferation.
6. When was the time of baseline GFR in Table 1? Normally, the baseline GFR should be addressed at the time of steady state.
Baseline eGFR was calculated based on the value of the first serum creatinine obtained in the first 48h after KT.
7. Currently, the HLA-DSA seems to be an established risk factor for graft deterioration, thus, the authors should mention as limitation of this study.
As we mentioned in the method section (Rows 06-08), preformed and de novo HLA-DSA were both evaluated. In the statement “In our cohort, none of the patients had performed HLA-DSA and only one patient, from the AT1R-Ab negative group, developed de novo HLA-DSA-Ab” from the discussion section, an error occurred due to autocorrect, which leads to a misunderstanding. The correct form is “preformed HLA-DSA” and not “performed HLA-DSA”. The mistake has been corrected. In this context, there is no need for a limitation mention.
8. The authors should provide some evidence from their study to support why the ER-TAC regimen showed more decreased eGFR compared with IR-TAC. For example, trough TAC levels, TAC IPV between groups, etc.
We found that eGFR was significantly more decreased in IR-TAC than in ER-TAC group.
We introduced statistical data and modified the following statement:’’ No significant difference regarding mean TAC trough level and IPV according to AT1R-Ab and TAC type has been observed’’:
No significant difference regarding mean TAC trough level (AT1R-Ab positive group, IR-TAC: 10.2± 0.1 ng/ml vs ER-TAC: 9.3± 1.9 ng/ml, p=0.56; AT1R-Ab negative group, IR-TAC: 8.5± 1.1 ng/ml vs 8.4± 0.7 ng/ml, p= 0.88) and median TAC IPV [AT1R-Ab positive group, IR-TAC: 19.7% (12.2- 35.5) vs ER-TAC: 15.2 % (3.4- 15.2%), p= 0.48; AT1R-Ab negative group, IR-TAC: 13.9% (7.1- 21) vs ER-TAC: 12.1 (3- 22.7), p= 0.63)] according to AT1R-Ab and TAC type has been observed.(Rows 279-283).
Even though mean TAC trough level and median TAC IPV were higher in patients with AT1R-Ab treated with IR-TAC than in those treated with ER-TAC, there was no significant difference, possibly influenced by the number of patients in the two groups (2 vs 10).
9.There are some typing errors, e.g, area (L183)
Word “aria” was corrected to “area”.
10. Please do recheck and complete Ref. #33.
Reference no. 33 (now 34) – Saengram W. et al. was checked and completed.
Round 2
Reviewer 1 Report
The statistical method for the multivariate analysis is not acceptable. The number of events are only 12. This multivariate analysis does not show precise results.
Reviewer 2 Report
The development of non-HLA antibodies (specifically AT1R-Ab) and their risk factors is doubtless a subject of interest. The authors have adequately addressed my previous concerns and substantially improved the final version of the manuscript.